# Design and Validation of a Test to Evaluate the Execution Time and Decision-Making in Technical–Tactical Football Actions (Passing and Driving)

**DOI:** 10.3390/bs13020101

**Published:** 2023-01-26

**Authors:** Guillermo Andres Calle-Jaramillo, Enoc Valentin Gonzalez-Palacio, Lewis Adrian Perez-Mendez, Andres Rojas-Jaramillo, Jose Antonio Gonzalez-Jurado

**Affiliations:** 1Escuela de Doctorado Universidad Pablo de Olavide (EDUPO), Universidad Pablo de Olavide (UPO), 41013 Sevilla, Spain; 2Grupo de Investigación Cultura Somática del Instituto Universitario de Educación Física (IUEF), Universidad de Antioquia (U. de A.), Medellín 050034, Colombia; 3Liga Antioqueña de Voleybol, Medellín 050034, Colombia; 4Grupo de Investigación de Ciencias Aplicadas a la Actividad Física y el Deporte (GRICAFDE), del Instituto Universitario de Educación Física (IUEF), Universidad de Antioquia (U. de A.), Medellín 050034, Colombia; 5Centro de Investigación en Rendimiento Físico y Deportivo, Universidad Pablo de Olavide (UPO), 41001 Sevilla, Spain

**Keywords:** outcome assessment, stroop task, cognition, reaction time, football

## Abstract

Reaction time and decision-making (DMA) in football have usually been evaluated using edited images or videos of game situations. The purpose of this research is to design and validate a test that simultaneously evaluates execution time (ET) and decision-making (DMA) in the subcategories of type of action (TA) and direction of movement (DM). Methodology: A quantitative, cross-sectional, and descriptive study of 30 young players. A total of 32 stimuli were programmed, corresponding to 64 responses, from which the total index (TI) was obtained from the division between DMA and ET. Results: The content validity index (CVI = 0.78) showed a high degree of consensus among experts. In the validation process, the intraclass correlation coefficient (ICC) was used to assess intraclass and interobserver reliability, and a moderate level of agreement was found between subjects for the TA (ICC = 0.593) and ET (ICC = 0.602) and a moderate high level of concordance for DM (ICC = 0.804) and TI (ICC = 0.855). Regarding interobserver reliability, an excellent level of agreement was found for all variables: TA (ICC = 0.998), DM (ICC = 0.998), ET (ICC = 1.000), and TI (ICC = 1.000). For the relationship between intraobserver and interobserver variables, statistical significance was established as *p* < 0.01. Finally, the intraobserver ETM (5.40%) and interobserver ETM (0.42%) was low compared with the reference value (5.9%). Conclusion: The designed test meets the validity criteria since the variables show sufficient intraclass reliability (test–retest) and reliability among observers.

## 1. Introduction

Reaction time is very important in sports performance and refers specifically to the time lapse between the appearance of the stimulus and the onset of a motor response [1]. Therefore, measuring the reaction time can become an effective strategy to determine the performance level of a player based on their decision-making (DMA).

The visual reaction time can be broken down into two moments: The first is the period from the appearance of a stimulus and saccade onset, which has been recorded as lasting between 180 and 300 ms [2]. The second moment is the duration of the nerve impulse from the sensory organ to the motor plate, which was reported to range from 180 to 200 ms [3]. In later studies, similar reference values have been reported; for example, Nakamoto and Mori [4] evaluated 20 basketball players and 24 university baseball players and found average latency times of 231 and 233 ms, respectively. In a study designed by Heilmann, with a sample of 68 young soccer players, an average response time of 315 ms (in the Cued Go/NoGo task), 446 ms (in the Flanker task), and 764 ms (3-back task) were reported [5], since these paradigms generate more complex cognitive demands.

In addition to the above, it is necessary to consider another moment, namely the decision time, which refers to the time it takes for decision-making (DMA) derived from the characteristic of the stimulus, which increases the total visual reaction time. Therefore, the instrument proposed in this text, given its characteristics, is inscribed in what Bonnet [3] calls the complex reaction time since it must discriminate the direction of the stimulus to respond to according to the stipulations. This was tested by Torres-Tejada et al. [6], who, in the same experiment, used stimuli that each required their own response (arrows in four directions) and combined them with stimuli for which responses should be inhibited.

In the same way, it is essential to clarify that cognitive flexibility (including creatively thinking “outside the box,” seeing anything from different perspectives, and quickly and flexibly adapting to changed circumstances) [7] is implicit in decision-making (DMA), defined as an action choice, and it is an outcome that can be observed as a motor or verbal response [8]. Thus, a requirement of this work is that the concept of spatial Stroop or arrow tasks be addressed because they generate conflict and interference and because the response depends on the direction of the arrow (choice of the relevant stimulus) and not on its location (suppression of the irrelevant stimulus) [9]. This needs to be explained because the anterior cingulate cortex and other prefrontal areas that comprise the anterior attentional network are activated when the number of elements to be selected increases and when the conflicting blocks of the Stroop task appear [10]. This originates from a change in the rules of a paradigm that requires participants to discriminate the direction of the arrows, which in turn may appear in a congruent or incongruent position with respect to that direction [11].

In an attempt to get closer to real situations of play in sports, an explicit task-switching procedure is used, which is a method to study the change of tasks [12,13,14] generating modifications in behaviors and thoughts and using executive control processes, such as working memory, inhibition, and cognitive flexibility [7]. When two consecutive trials are different because there is alternation in tasks (signal change and task change), the first produces an inertia effect on the second, and the conflict to change the response increases. This cost of change can be explained by the difficulty of active and voluntary inhibition to disengage from the previous stimulus [15,16]. Executive control refers to the processes by which the mind controls itself to choose and monitor behavior, taking into account the consequences of this and the possibility of disengaging if deemed necessary [17,18,19]. In this way, multiple concurrent tasks originated by perceptual, cognitive, and motor processes [18] modulate the reaction time.

Cognitive flexibility is interwoven with decision-making (DMA) to change the direction of movement (DM) and the type of action (TA) present in interactive sports (team/ball games) because, in addition to physical technique and equipment, there is also dependence on cognitive skills to obtain results [20,21]. For example, football is an oppositional team sport where players must quickly assess information about the ball regarding their teammates, opponents, and location on the field before considering the appropriate action based on their skills, the coach’s instructions, and the current situation of the match [22]. When it comes to shooting, passing, or dribbling, the football player needs exceptional visual perception to register opponents and teammates approaching as well as quick decision-making (DMA) and inhibitory control to change the motor plan if a defender prevents the execution of the current one [23].

In this sense, decision-making (DMA) was evaluated using videos of offensive actions from official matches (11 vs.11) of the main European football leagues (Spain, England, Italy, and Germany). The participants observed a game situation during a limited time (5–13 sec.) and had to anticipate the following technical action. The evaluators registered the correct answers and their respective delay times [24]. In a similar study, the decision-making (DMA) of football players of various ages (U16, U18, and U23) was evaluated using edited video clips from the Premier League. Three UEFA-licensed coaches rated the decision-making (DMA) independently, and only assessments where two or all three raters agreed were used [25]. Other studies used video sequences of Bundesliga matches to assess creativity in decision-making (DMA). The football players were asked to confirm, justify, and explain their answers verbally. Three independent evaluators (football coaches qualified by UEFA) judged the originality of the solutions given by the participants for each game situation raised, using a Likert-type scale ranging from 1 (not original at all) to 5 (very original) [26,27]. In addition, decision-making (DMA) was evaluated using global motor responses to offensive game situations lasting 3 s. The participants should select the most precise motor response in the least amount of time. Reaction time was measured from the time the video stopped until the player’s lower extremity touched the ball. The selection of the correct responses determined the accuracy of the motor response. The assessors were a group of professionals composed of six UEFA-licensed coaches who used a Likert scale to assess the players [28]. Similar research, but aimed at capturing brain activity, revealed changes in electroencephalogram (EEG) and functional magnetic resonance imaging (fMRI) recordings when participants imagined creative or conventional movements in decision-making (DMA) situations in football after watching video clips. The scenes lasted between 2 and 12 s, and the time for the imagination of the movement could not exceed 15 s; once they imagined the movement, they stopped it [29,30].

Consequently, the purpose of this research is to design and validate a feasible instrument based on paradigms with scientific evidence used in neuropsychology and that, in laboratory situations, allows for the determination of the performance level of a player, obtaining accurate and relevant information on the execution times (ET) and decision-making (DMA) present during the execution of motor actions in football (specifically passing and driving). This test evaluates decision-making (DMA) using technical and tactical actions typical of football, and although it is a laboratory test, it is executed in an open field, unlike some antecedents, which evaluate decision-making (DMA) in an enclosed space restricting motor response. When designing the study, it was hypothesized that the test would meet the requirements of validity and reliability due to a design based on validated paradigms of cognitive neuroscience, which produce conflicts, interferences, and disturbances in the executive functions that underlie decision-making (DMA) and that modulate decision time. The validation of this test will allow sports science professionals to evaluate more objectively the technical-tactical performance of offensive football actions. The test, based on its characteristics, is called the Stroop Task Football Test (STFT) or “*Test de Calle*”.

## 2. Materials and Methods

This was a quantitative, cross-sectional, and descriptive study [31]. The dependent variables included the total index (TI), execution time (ET), and decision-making (DMA). Decision-making (DMA) included the subcategories: type of action (TA) and direction of movement (DM).

The sample/subject participants consisted of 30 male footballers in the under-20 category and belonging to two teams in the lower divisions of Colombian professional football: Envigado Fútbol Club (18 players) and Leones Fútbol Club (12 players), who placed second and third, respectively, in the Antioquia Football League category 1C tournament. The characteristics of the sample (average ± DS) were as follows: age (18.43 ± 0.935 years), height (176.78 ± 2.74 cm), weight (71.44 ± 3.13 kg), and time spent in the club: (8.33 ± 1.38 years).

Since the study aimed to establish the agreement between two measures and thus establish their reliability, the sample size to determine the magnitude of the intraclass correlation coefficient (ICC) was calculated with a reliability of 95% (*Z* = 1.96) and a statistical significance of 5%, considering a sample estimate of 0.96 and a confidence interval of 95%, indicating an almost perfect degree of agreement [32]. Therefore, although a minimum sample of 25 subjects was determined, the validation of this test was finally done with 30 subjects, a value that is analogous to other similar studies; for example, the study conducted by Moal et al. [33] used two of its groups with 22 and 21 people; for another validation carried out by Machado et al. [34], subgroups of 15 people were used, and the study by Bian et al. [35] performed validation with 15 subjects. It should be noted that the sample used is professional athletes, so the expansion of this is difficult in the sports context. The formula for the calculation of the sample was as follows:
$$n={\left[\frac{Z_{1-\alpha /2}}{\left(0.5ln\frac{1+r}{1-r}\right)-\left(0.5ln\frac{1+r-ICC_{h}}{1-r+ICC_{h}}\right)}\right]}^{2}+3$$

There were seven days between the tests and the retests. Both teams were at a time in the season when they had similar levels of demand due to the number of competitions and the number of training sessions in the respective year.

The tests and retests were performed at the same time of the day. The tests were performed 48 h after a workout. The practice was explained before it was carried out, with feedback given in response to any questions. Prior to tests and retests, participants performed activities typical of training, taking rest periods of greater than five minutes after such activities.

**Environmental characteristics:** The test was performed between 8:00 and 10:00 a.m. at a temperature between 18 and 25 degrees Celsius on football fields with synthetic grass. An official FIFA Golty^®^ ball was used.

**Inclusion criteria:** Signed informed consent, performing both tests (test and retest), and having attended all training in the immediately preceding weekly cycle.

**Exclusion criteria:** Having consumed caffeine or any other stimulant substance that day, diagnosis of any psychiatric or neurological problems, and injuries or physical discomfort at the time of the test.

Of a total of 46 players (see Figure 1) belonging to the 2 teams, 10 players did not meet the inclusion criteria, and 6 players were excluded for having presented injuries or physical discomfort at the time of the test (follow-up). Finally, in the analysis of the information, the records of 30 players were taken into account.

**Ethical considerations:** We refer to the ethical provisions of the Declaration of Helsinki.

### Design of a Test to Evaluate Execution Time (ET) and Decision-Making (DMA) in Technical–Tactical Football Actions

The test was built from paradigms used by cognitive neuroscience to assess reaction times through motor responses. The paradigms applied are Complex Reaction Time [3], Choice Task Paradigm [6], Spatial Stroop [9], and Explicit Task-Switching Procedure [12,13,14,36]. The delay time is an indicator of the cognitive performance of the subjects since the instructions of these paradigms cause conflicts, interferences, and disturbances in the effectiveness of task switching and in cognitive flexibility.

In this study, the reaction time in each trial is not recorded but rather the sum of all the trials performed by each subject. So, the final result will be the total time it took to execute the test. This time is equal to the execution time (ET).

Likewise, the subject had to make decisions regarding the direction of movement (DM) and the type of action (TA) to be executed (driving or passing); that is, the correct or incorrect decision depended on whether the player executed the corresponding movement and in the indicated direction. The direction was determined by the direction rather than the location of the arrows. In this way, the trials can be congruent or incongruent with respect to the location and direction of the stimuli. In congruent trials, the location and direction of the arrow coincide I↖I↗I↙I↘I, whereas, in incongruent trials, there is no such correspondence I↗I↖I↘I↙I [9,37].

Expert athletes have a greater capacity to perceive relevant indices or pre-indices to anticipate the next action [38,39,40]. Consequently, the test design may favor a footballer who identifies the location of the arrows (up or down) as a pre-index to discard half of the answers related to the direction of movement (DM).

Between trials, the task may be changed [36], for example, regarding the color of the arrows in determining the type of action (TA) to be executed, with red for passing and green for driving. The colors red and green were chosen on the basis of the trichromatic theory formulated by Young and Von Helmholtz, which proposes the existence of three cones (photoreceptor cells of the retina) that have different sensitivities to specific wavelengths (C cones, blue—short wave; M cones, green—medium wave; L cones, red—long wave). In addition, the theory of opponent processes proposed by Ewald Hering says that there are three pairs of colors that are impossible to combine: red and green, black and white, and blue and yellow [41]. In this sense, and considering the two theories, we chose the colors red and green for the contrast they generate, thus avoiding unnecessary conflicts originating in the signals of the test.

The blocks were composed of 32 trials, where, for 32 stimuli, there are 64 possible responses since the type of action (TA), and the direction of movement (DM) (i.e., 2 items) are evaluated in each response. The stimuli appeared randomly in one of the four quadrants in a congruent or incongruous manner and with red or green arrows. That is, the distribution of stimuli was divided into 16 stimuli for passing and 16 stimuli for driving. Each action had 8 congruent trials and 8 incongruent trials (for each type of trial, there were 2 stimuli per quadrant) (see Figure 2).

Regarding the direction, movements were executed on a surface demarcated by four arrows, where their intersection was the starting point. From the center to the periphery (tip of the arrows), there was a distance of 2.40 m, and there was also a wall that measured 0.60 m for the rebound of the ball in the trials related to the pass. Likewise, a box that had 1.20 m of the distance between each side was demarcated, and the participants executing a pass outside this box were penalized. Meanwhile, driving required the player to exit and re-enter the frame. In addition, the box corresponds to the reference for the evaluator to show the next signal pressing the spacebar when the ball crosses the borders (see Figure 3).

The spatial Stroop effect corresponds to incongruent trials, where the direction of the arrows does not coincide with their location. The tendency is to respond toward the location of the stimulus rather than its direction; therefore, the participant must suppress that tendency and always respond toward the direction (see Figure 2).

The computer screen where the stimuli originated was located at a distance of 4.80 m and at the height of 60 cm between the screen and the center of the “X” there were balls to be used by the participant when, due to a lack of precision, the pass did not hit the rebound walls (see Figure 3).

In conclusion, the successes achieved by good decision-making (DMA) related to the type of action (TA) and the direction of movement (DM) were scored as (1), and errors for the opposite reasons were scored as (0) (see Figure 4). It should be noted that a lack of precision was also penalized and qualified as an error, like executing the pass out of the box.

**Technological equipment used for the capture and analysis of videos and information:** ASUS TUF Gaming F15 FX506LI-HN011 Laptop Specifications. Operating System: Windows 10 Home—ASUS.

**Program design and description of the object system:** the Stroop Task Football Test (STFT) system is directional stimulus software that is intended to randomly display 1 of a possible 32 variations of a stimulus on a screen. The system comprises an initial input screen to start the stimulus and an output screen with the option to repeat the stimulation again. The system operator fulfills the function of passing the stimuli with the signal of the spacebar. Finally, the system saves the partial times in which each stimulus change was recorded as well as the total time of the test.

**Scope:** the system allows for showing a stimulation cycle randomly without following any type of statistical distribution at the time of randomization. Additionally, the date and time are recorded for each instance in which the stimulus is changed, and the total of each cycle is taken as the total of the session.

**Functionality:** the system was developed and tested on Windows 10.

**Construction:** the stimuli were built in Microsoft PowerPoint with the agreed parameters, generating images of 1305 × 735 pixels. The system was built in Python 3.9 (Amsterdam, The Netherlands), using scientific libraries such as Numpy, Pandas, and Datetime for the management of images within the environment. Finally, the graphics component was developed with the Pygame book [42].

Diagram flow and system map: the diagram presented is the one proposed for the management of the system (see Figure 5).

**Use:** The executable file (STFT.exe) is available to download in the Appendix A.

The executable system program is hosted in the generated folder. Moving the executable from that folder should be avoided; failing that, if it needs to be moved, a shortcut should be created.Double-click on the executable program to open the initial screen (see Figure 6).Pressing the spacebar will start the stimulus, as shown in Figure 7.If the block is finished, it is possible to make the decision to exit the stimulus or to start it again. The ESC key is pressed to exit, and the spacebar is pressed to go back to the stimulation images. “The cycle time was 0:00:52, equivalent to the block time. The blocks were composed of 32 trials, where, for 32 stimuli, there are 64 possible responses, since the type of action (TA) and the direction of movement (DM) (i.e., 2 items) are evaluated in each response” (see Figure 8).After finishing each circuit, a folder will be generated in the folder where the executable is located, with the partial times corresponding to the duration between the stimuli caused by the pressure of the spacebar. The txt file contains the recorded times, named according to the date and time they were generated (see Figure 9).

**Information Collection**: the test was recorded in audio and video with the ASUS computer, and the blocks were then analyzed as equivalent to 32 trials per player. Parallel to the filming, the screen that emitted the stimuli was recorded at random times using the OBS STUDIO program. Once the information was captured, each filmed trial was compared with its corresponding stimulus from the same screen, digitizing the evaluation in an Excel sheet. Finally, the data were transcribed in the SPSS Statistical Software version 23 (see Figure 10).

## 3. Results

The construction of the instrument was based on the measurement of the test execution time (ET) and the evaluation of decision-making (DMA) in two subcategories: the type of action (TA) and the direction of movement (DM). In consideration of this, and understanding that the test had three variables, nine experts were invited, of which six answered questions related to the assessment of each variable (essential and non-essential), presenting content validity ratio (CVR) well above of the reference value (0.5823). The variable direction of movement (DM) presented a total consensus, while in the variables type of action (TA) and execution time (ET) the consensus was in four experts (14 approvals out of a total of 18 possible approvals). Consequently, the content validity index (CVI) revealed a high degree of consensus among the judges who evaluated the test [43] (see Table 1).

On the other hand, to determine the reliability of the test and its components, the criteria of logical validity proposed by Safrit [44] were used, for which a test–retest was carried out with one week between the two, and an independent evaluation was conducted by two experts in the field. It should be clarified that before carrying out the statistical procedures, the normality of the variables to be compared was evaluated using the Shapiro–Wilk test, according to which all variables were found to be normally distributed (*p* > 0.05)

### 3.1. Intraclass Reliability

When evaluating the intraclass reliability of the procedure (test–retest) in terms of the test variables, we found a moderate level of agreement for the variable’s type of action (TA) and execution time (ET), with ICC values of (0.593) and (0.602), respectively, and a statistical significance of (*p* < 0.01) (see Table 2 and Table 3). In addition, in the variable direction of movement (DM) (ICC = 0.804) and the total index (TI) of the test (0.855), the level of agreement was good, with a statistical significance of (*p* < 0.01) for both (see Table 4 and Table 5). This indicates that the test has very good stability over time.

### 3.2. Interobserver Reliability

Likewise, for the interobserver reliability, there was an excellent level of agreement for the variables type of action (TA) (ICC = 0.998) and direction of movement (DM) (ICC = 0.998), with statistical significance for the two variables of *p* < 0.01 (see Table 6 and Table 7). Likewise, the reliability was excellent (ICC = 1000) for the variable execution time (ET) and the total index (TI) variable of the test. These two variables had a statistical significance of *p* < 0.01 (see Table 8 and Table 9). In other words, there was essentially total agreement among the experts who evaluated the test. Thus, the relationship between the intraobserver and interobserver variables showed good concordance at the 95% confidence interval.

Finally, the technical error of intrasubject measurement (test–retest) was 5.40%, and the technical error of interobserver measurement was 0.42%. These figures are below the reference values (5.9%) in statistical terms; that is, the error between the measurements is low, and, therefore, they are recommended and considered to be of good quality.


**Technical error of measurement (TEM) INTER (observer 1–observer 2);**
Absolut TEM = √∑D²/2*n* = 0.22; Relative TEM = ETM/Mv = 0.42%;



**Technical error of measurement (TEM) INTRA (Test–Retest);**
Absolut TEM = √∑D²/2*n* = 2.79; Relative TEM = ETM/Mv = 5.40%.


## 4. Discussion

In the present study, paradigms of cognitive neuroscience were used to design and validate a laboratory test based on visual stimuli in which, based on its instructions, the execution time (ET) (equivalent to the sum of the reaction times of each trial) and decision- making (DMA) are evaluated in terms of motor responses of the lower limbs that execute two relevant and necessary tactical, technical actions in football (passing and driving). The main contribution of this research is the development of a validated test that will provide objective information on the cognitive-motor performance of a footballer.

Physiological-based procedures do not appear to be sensitive enough to identify professional athletes from semi-professionals [45]; it is necessary to have a multivariate approach [46]. According to cognitive neuroscience, executive functions (working memory, inhibition, and cognitive flexibility) [47,48] that underpin decision-making (DMA) are required in non-everyday situations [7], such as football, to achieve success. The efficiency in the application of cognitive functions to sports is one of the factors that allows expert athletes to be superior to novice and non-athlete athletes [49]. Thus, players considered to be elite can obtain better results in neuropsychological tests that evaluate variables such as attention, inhibitory control, or cognitive flexibility with respect to amateur players or sedentary subjects [49,50,51].

Football is characterized as an open-skill sport because it creates changing tactical circumstances that force players to make rapid and effective judgments in a dynamic and unexpected environment [52,53], resulting in significant favorable impacts on executive functions. A football game puts a high strain on executive processes because athletes must adjust to changing situations (cognitive flexibility), stop actions in the near term (inhibition), or quickly recall essential information connected to the current situation (working memory) [54].

Several laboratory methods study perceptual and cognitive skills in football [22,55], examining the number of central and peripheral fixations that the eye has on the ball and off it [56,57] or evaluating the best-attacking option (passing, shooting, dribbling, or running) while the participant observes images on a screen from the first-person perspective [58,59,60]. Likewise, the athlete is evaluated while watching edited footage of the sport they practice and are asked what the best decision would be regarding a certain move [57]. In addition, the short-term memory of the player is evaluated, showing sequences of actions for short periods of time so that, seconds later, they can try to recall, as accurately as possible, the details of the scenes seen in the different video clips [61]. In children’s and youth football, other measurement instruments have been used to evaluate decision-making (DMA) on the field in reduced games (GPAI and GPET) [62,63].

The results of this study that measures objective parameters (execution time and success in decision-making) differ from others that assess decision-making (DMA) and reaction times in football using edited videos of offensive situations obtained from official matches. Participants observed each scene for a few seconds before stopping and requesting a verbal and/or motor answer as quickly as possible [24,25,26,27,28]. The decision-making (DMA) was evaluated by a panel of expert coaches mainly applying Likert-type scales.

Likewise, similar researchers analyzed brain activity using electroencephalogram (EEG) and functional magnetic resonance imaging (fMRI) when participants imagined creative movements or conventional movements in decision-making (DMA) situations after presenting video clips [29,30].

On the other hand, there have been neuropsychology studies on cognitive functions using paradigms designed to quantify reaction times via digital motor responses. Consequently, the design of the present test, which uses foot motor responses, is based on the following neuropsychological tests: Complex Reaction Time [3], Choice Task Paradigm [6], Spatial Stroop [9], and Explicit Task-Switching Procedure [12,13,14,36].

Studies cited in this paper use complex tests to assess decision-making (DMA) and response times [24,25,26,27,28]. However, the novelty of this test is that its metrics, decision time and decision-making (DMA), are easy to evaluate since the paradigms used have been previously validated by cognitive neuroscience. In addition, these paradigms intentionally add another time (decision time) to the reaction time due to the conflicts, interferences, and disturbances present at the time of decision-making (DMA).

Meanwhile, the intrasubject reliability was similar to that found in the validation of the Loughborough Soccer Passing Test [33], where the relationship between two attempts separated by a week was very high (*r* = 0.73–0.96).

Regarding the interobserver reliability, this tool presented moderate and high values, which is consistent with other field tests that evaluate decision-making (DMA) in the tactical elements of football, for example, the FUT-SAT [64], presenting interobserver reliability values (0.79 and 0.87). In the same way, the reports of the current study agree with the results of another tactical test, such as the TacticUP [34], where the test–retest reliability was evaluated using the kappa index, yielding nearly perfect values (0.622 and 1.0). In this test, the ICC was applied, which was moderate to high (ICC = 0.804–0.0855).

In summary, decision-making (DMA) in football has been evaluated in multiple ways, including the assessment of the technical elements present in it, as well as its participation in the review of the tactical elements present in the internal logic of this sport. The most frequent method used has been the analysis of videos by experts–coaches. This study evaluated decision-making (DMA) in laboratory conditions, a fact that facilitated validation based on content and reliability, interobservers, and test–retest criteria. However, the test used in this study lacks ecological validity, which limits the evaluation of decision-making (DMA) in a real-world setting (limitation of the study). Another limitation of this tool compared to other studies is that it does not use images or videos of real sports situations but rather shows non-specific visual stimuli. The objective of this tool is to evaluate the accuracy and speed of decision-making (DMA) in situations that are as objective as possible.

Football requires choosing the best option in the shortest possible time, so this study, in addition to determining the right decision, also recognizes the decision time as a component of the reaction time and that its sum is related to the execution time (ET) of the test. Therefore, having tools to evaluate efficiency in decision-making (DMA) would be very useful for the selection of players, planning training, determining the approach to the competition, fatigue detection, load monitoring, and injury monitoring.

In this way, the intention is to provide sports science with evidence acquired in cognitive neuroscience, seeking to increasingly reduce the gap between the areas of knowledge. What to do, when, where, and how to do it are the questions our brains solve when movement is intelligently executed.

It is proposed that future research should use this test to evaluate performance levels among novices and experts, in situations of fatigue, during the season, and due to musculoskeletal injuries. The obtained data may be compared with data from other psychomotor and neuropsychological tests, from other field tests, and with the subjective perceptions of coaches as one more tool to consider. It can also be a part of a series of tests used by sports professionals to monitor the performance of a player in different situations and moments of the season.

### Practical Considerations

This test is a tool that allows for evaluating cognitive-motor performance in footballers.It is an easy-to-use tool that is at the hand of any coach or sports scientist.It is a tool with high reliability among observers and high stability over time.In a tool that brings neurosciences closer to the sports field.

## 5. Conclusions

The results show that the developed test is stable over time and can be used by any evaluator with prior knowledge since the level of agreement on the part of the observers showed good consistency. Therefore, the designed test meets the criteria for validity since the variables show intraclass reliability (test–retest) and reliability among observers. The variables were correlated with each other and obtained a result indicative of an athlete’s performance. In this way, the Total Index (TI) or cognitive-motor index (CMI) was obtained from the sum of the successful attempts (a maximum of 64: 32 types of action (TA) and 32 directions of movement (DM) corresponding to 32 trials; each trial allowed for qualifying both the action and the direction) divided by the execution time (ET) of the test or the block (each block contained 32 trials).

## Figures and Tables

**Figure 1 behavsci-13-00101-f001:**
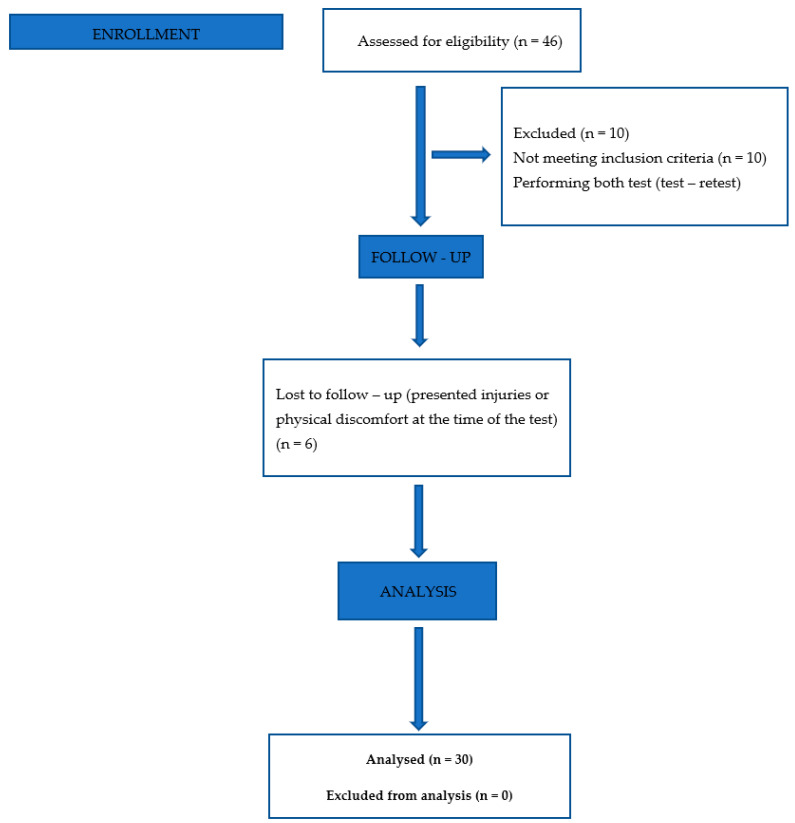
Flowchart: inclusion/exclusion criteria to obtain the sample of test participants.

**Figure 2 behavsci-13-00101-f002:**
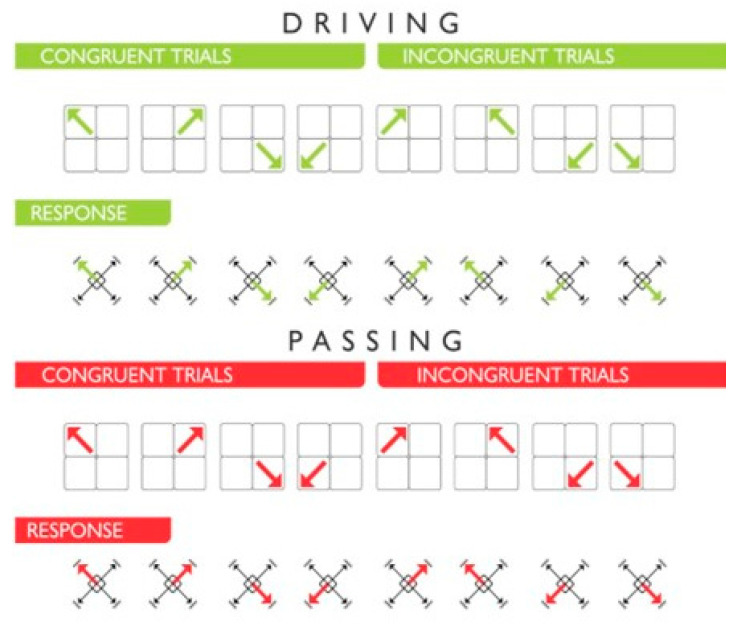
Scheme of stimuli and responses. The stimuli appeared in one of the four quadrants in a congruent or incongruous manner and with red or green arrows. The response appeared in one of the four directions.

**Figure 3 behavsci-13-00101-f003:**
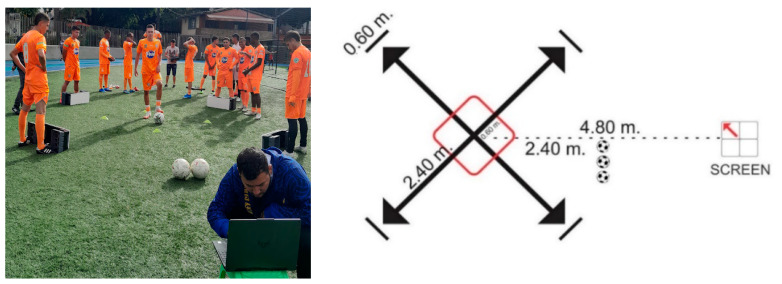
Characteristics of the surface where the test was executed. Regarding the direction, movements were executed on a surface demarcated by four arrows, where their intersection was the starting point. From the center to the periphery (tip of the arrows), there was a distance of 2.40 m, and there was also a wall that measured 0.60 mts for the rebound of the ball in the trials related to the pass. Likewise, a box that had 1.20 m of the distance between each side was demarcated. The computer screen where the stimuli originated was located at a distance of 4.80 m and at the height of 60 cm between the screen and the center of the “X” there were balls to be used by the participant when, due to a lack of precision, the pass did not hit the rebound walls.

**Figure 4 behavsci-13-00101-f004:**
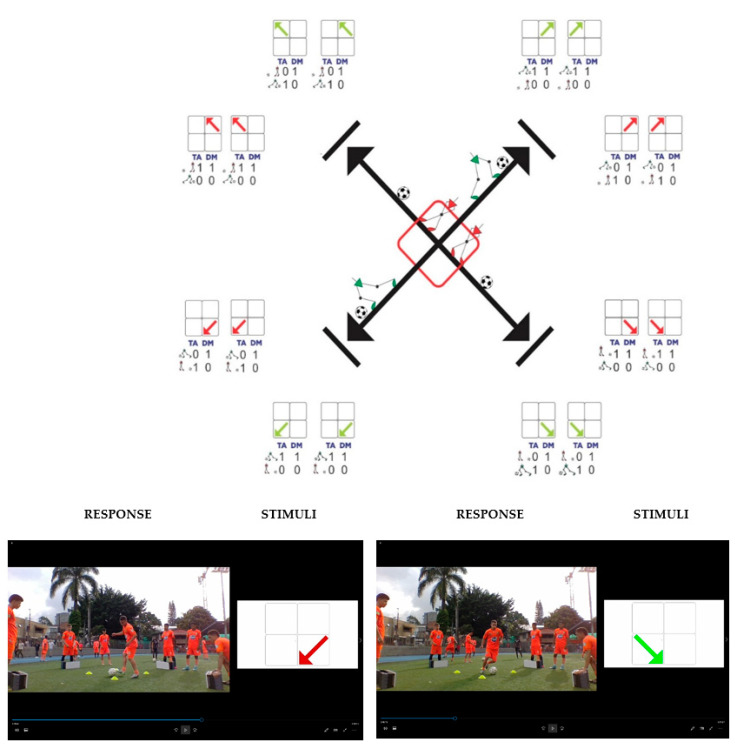
Examples of test grading. Good decision-making (DMA) related to the type of action (TA) and the direction of movement (DM) were scored as (1), and errors for the opposite reasons were scored as (0). It should be noted that a lack of precision was also penalized and qualified as an error, like executing the pass out of the box.

**Figure 5 behavsci-13-00101-f005:**
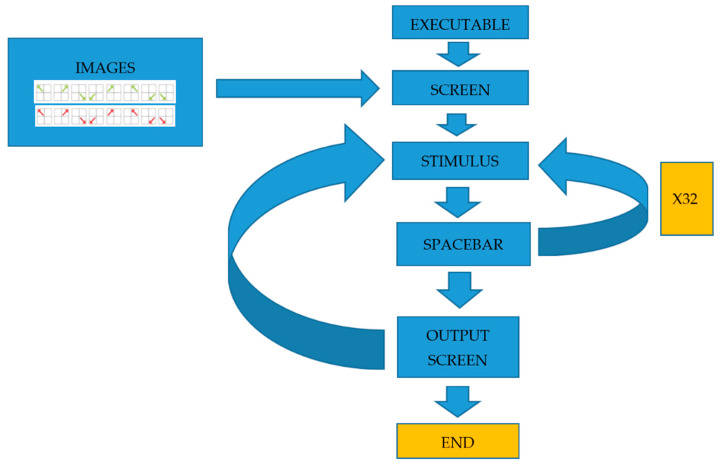
System flowchart.

**Figure 6 behavsci-13-00101-f006:**
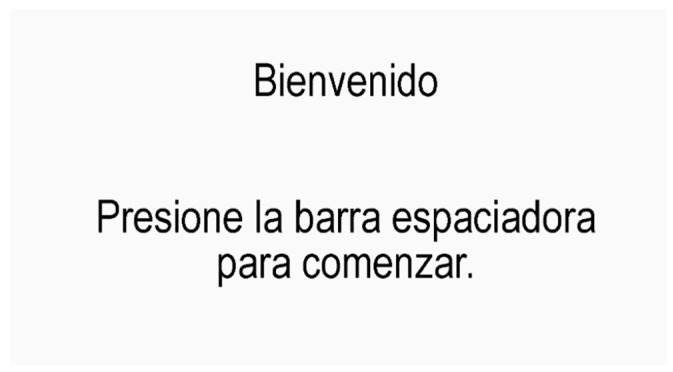
Initial screen.

**Figure 7 behavsci-13-00101-f007:**
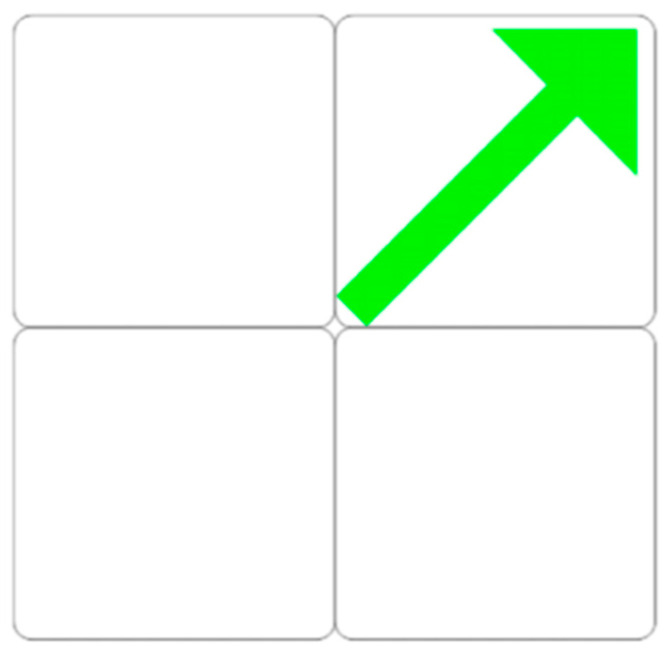
Congruent or incongruent stimulus.

**Figure 8 behavsci-13-00101-f008:**
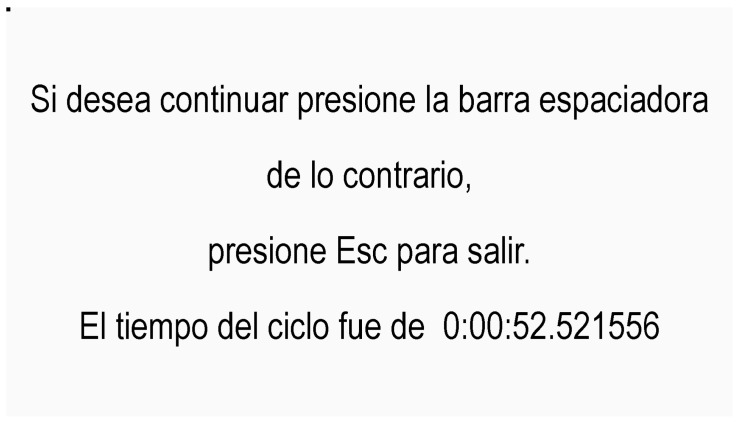
Information about block termination.

**Figure 9 behavsci-13-00101-f009:**
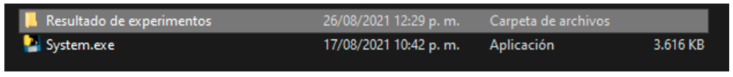
File with the partial times recorded.

**Figure 10 behavsci-13-00101-f010:**
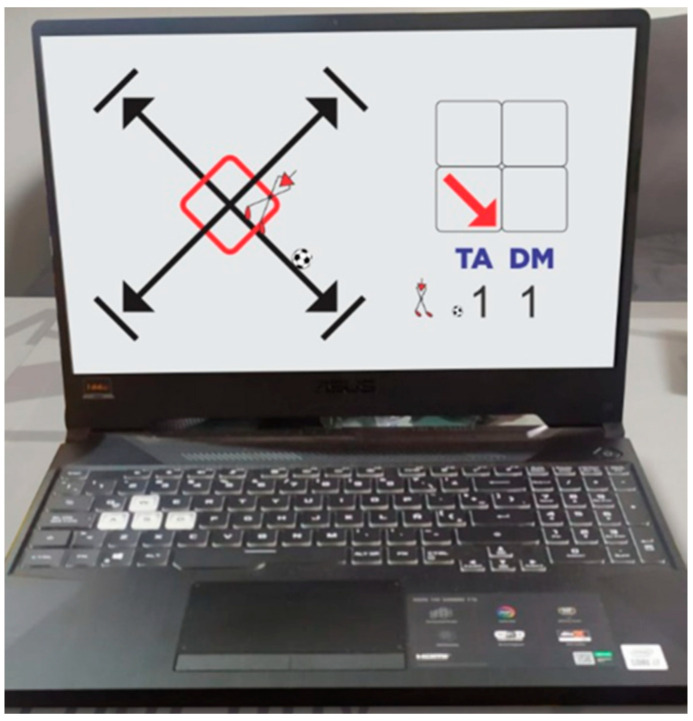
Screen with the stimuli and the player for the collection of information. Each filmed test was compared with its corresponding stimuli from the same screen.

**Table 1 behavsci-13-00101-t001:** CVR and CVI. The content validity index (CVI) revealed a high degree of consensus among the judges who evaluated the test.

Variable	Expert Assessment		CVR	CVI
	C	No—C		
Direction of movement	6	0	1	
Type of action	4	2	0.67	
Execution time	4	2	0.67	
				0.78

CVI: Content validity index; CVR: Content validity ratio; C: Consensus; No—C: No consensus.

**Table 2 behavsci-13-00101-t002:** Values in the ICC in the test–retest assessment of Type of Action (TA) and average measurement.

Variable			Average Measurements				
TA	Mean	SD	ICC	95% (CI)		F	*p* Value
				LL	UL		
Test	30.90	2.218	0.593	0.152	0.805	2.455	0.008
Retest	31.73	0.691					

ICC: Intraclass correlation coefficient; TA: Type of Action; SD: Standard Deviation; LI: Lower Limit; UL: Upper Limit.

**Table 3 behavsci-13-00101-t003:** Values in the ICC in the test–retest assessment of Execution Time (ET) and average measurement.

Variable			Average Measurements				
ET	Mean	SD	ICC	95% (CI)		F	*p* Value
				LL	UL		
Test	1.1320	0.07364	0.602	0.172	0.810	2.515	0.007
Retest	1.1417	0.06417					

ICC: Intraclass correlation coefficient; ET: Execution Time; SD: Standard Deviation; LI: Lower Limit; UL: Upper Limit.

**Table 4 behavsci-13-00101-t004:** Values in the ICC in the test–retest assessment of Direction of Movement (DM) and average measurement.

Variable			Average Measurements				
DM	Mean	SD	ICC	95% (CI)		F	*p* Value
				LL	UL		
Test	26,97	5.216	0.804	0.593	0.907	5.112	0.000
Retest	27.80	4.147					

ICC: Intraclass correlation coefficient; DM: Direction of Movement; SD: Standard Deviation; LI: Lower Limit; UL: Upper Limit.

**Table 5 behavsci-13-00101-t005:** Values in the ICC in the test–retest assessment of Total Index (TI) and average measurement.

Variable			Average Measurements				
TI	Mean	SD	ICC	95% (CI)		F	*p* Value
				LL	UL		
Test	51.3880	6.83409	0.855	0.698	0.931	6.897	0.000
Retest	52.3251	5.17441					

ICC: Intraclass correlation coefficient; TI: Total Index; SD: Standard Deviation; LI: Lower Limit; UL: Upper Limit.

**Table 6 behavsci-13-00101-t006:** Values in the ICC in the interobserver assessment of Type of Action (TA) and average measurement.

Variable			Average Measurements				
TA	Mean	SD	ICC	95% (CI)		F	*p* Value
				LL	UL		
Observer 1	30.90	2.218	0.998	0.996	0.999	586.931	0.000
Observer 2	30.87	2.209					

ICC: Intraclass correlation coefficient; TA: Type of Action; SD: Standard Deviation; LI: Lower Limit; UL: Upper Limit.

**Table 7 behavsci-13-00101-t007:** Values in the ICC in the interobserver assessment of Direction of Movement (DM) and average measurement.

Variable			Average Measurements				
DM	Mean	SD	ICC	95% (CI)		F	*p* Value
				LL	UL		
Observer 1	26.97	5.216	0.998	0.997	0.999	655.648	0.000
Observer 2	27.00	5.246					

ICC: Intraclass correlation coefficient; DM: Direction of Movement; SD: Standard Deviation; LI: Lower Limit; UL: Upper Limit.

**Table 8 behavsci-13-00101-t008:** Values in the ICC in the interobserver assessment of Execution Time (ET) and average measurement.

Variable			Average Measurements				
ET	Mean	SD	ICC	95% (CI)		F	*p* Value
				LL	UL		
Observer 1	1.1320	0.07364	1.000	1.000	1.000		1.000
Observer 2	1.1320	0.07364					

ICC: Intraclass correlation coefficient; ET: Execution Time; SD: Standard Deviation; LI: Lower Limit; UL: Upper Limit.

**Table 9 behavsci-13-00101-t009:** Values in the ICC in the interobserver assessment of Total Index (TI) and average measurement.

Variable			Average Measurements				
TI	Mean	SD	ICC	95% (CI)		F	*p* Value
				LL	UL		
Observer 1	51.3880	6.83409	1.000	0.999	1.000	3955.445	0.000
Observer 2	51.3865	6.82670					

ICC: Intraclass correlation coefficient; TI: Total Index; SD: Standard Deviation; LI: Lower Limit; UL: Upper Limit.

## Data Availability

The data presented in this study are openly available at: https://docs.google.com/spreadsheets/d/1c3bZNClvy8TP7RBspwRI0z7R8-n5Wy5H/edit?usp=sharing&ouid=108646522251636979792&rtpof=true&sd=true (accessed on 28 December 2021).

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
