# Peer review of "Design and Validation of a Test to Evaluate the Execution Time and Decision-Making in Technical–Tactical Football Actions (Passing and Driving)"

_behavsci, 2023, doi:10.3390/bs13020101_

Round 1

Reviewer 1 Report (Previous Reviewer 1)

The authors have significantly worked on their manuscript, which has improved its overall quality. 

Author Response

Thank you very much.

Reviewer 2 Report (New Reviewer)

Dear authors, thank you for the opportunity to review the article titled “Design And Validation Of A Test To Evaluate The Execution Time And Decision-Making In Technical–Tactical Football Actions (Passing 4 And Driving)”.  the purpose of this research is to design and validate a test that simultaneously evaluates execution time (ET) and decision-making (DMA) in the subcate gories type of action (TA) and direction of movement (DM). The results of the validation process were positive and meet the validity criteria, as it presented variables with sufficient intraclass reliability and reliability between observers. The study is interesting and can provide important information about decision making and execution time. However, some points must be considered for publication.       

Introduction:

There are some major points that I believe are important for the authors to adjust in the text:

 The first concerns the specification of terms and ideas to facilitate the reader's understanding of the subject matter, for example, what is the difference between decision time and decision-making reaction time? Are they synonyms? This must be clear in the work since it implies one of the main metrics of the study. It is important to be faithful to the concept and expressions you use throughout the introduction.

 It is important to emphasize that there are tests that have already been used that evaluate decision-making in relation to action and direction (Roca et al., 2011, 2018; Roca & Williams, 2016; Vaeyens et al., 2007; Vítor de Assis et al. ., 2020; Williams & Ward, 2007), the main novelty of this protocol is the decision time metric, I believe that this should be clear in the study and mainly explicit the differential of the decision time metric in relation, for example to the work of (Machado & Teoldo, 2020).

1.     Machado, G., & Teoldo, I. (2020). TacticUP video test for soccer: Development and validation. Frontiers in Psychology, 11(1690). https://doi.org/10.3389/fpsyg.2020.01690

2.     Roca, A., Ford, P. R., McRobert, A. P., & Williams, A. M. (2011). Identifying the processes underpinning anticipation and decision-making in a dynamic time-constrained task. Cognitive Processing, 12(3), 301–310. https://doi.org/https://doi.org/10.1007/s10339-011-0392-1

3.     Roca, A., Ford, P. R., & Memmert, D. (2018). Creative decision making and visual search behavior in skilled soccer players. PLoS ONE, 13(7), 1–11. https://doi.org/. https://doi.org/10.1371/journal. pone.0199381

4.     Roca, A., & Williams, A. M. (2016). Expertise and the Interaction between Different Perceptual-Cognitive Skills: Implications for Testing and Training. Frontiers in Psychology, 7, 792. https://doi.org/10.3389/fpsyg.2016.00792

5.     Vaeyens, R., Lenoir, M., Williams, A. M., & Philippaerts, R. M. (2007). Mechanisms Underpinning Successful Decision Making in Skilled Youth Soccer Players: An Analysis of Visual Search Behaviors. Journal of Motor Behavior, 39(5), 395–408. https://doi.org/10.3200/JMBR.39.5.395-408

6.     Vítor de Assis, J., González-Víllora, S., Clemente, F. M., Cardoso, F., & Teoldo, I. (2020). Do youth soccer players with different tactical behaviour also perform differently in decision-making and visual search strategies? International Journal of Performance Analysis in Sport. https://doi.org/10.1080/24748668.2020.1838784

7.     Williams, A. M., & Ward, P. (2007). Anticipation and Decision Making in Sport. In G. Tenenbaum & R. C. Eklund (Eds.), Handbook of Sport Psychology (3rd ed., pp. 203–223). John Wiley & Sons.         

I think that the instrument primarily assesses decision-making within a technical context, however little is said about this in the introduction.

 Finally, I believe that the authors should also clearly define the justification for the study and add the hypotheses that were raised after the objectives.

 Some minor points:

P2 – L48 and 49 Bring evidence in football.

P2-L59 – I think it is important to familiarize the reader from other areas to bring some brief explanation about what cognitive flexibility is, or else direct to an article that explains what it is.

Materials and methods:

The efforts of the authors to validate a test of this size are commendable, in addition, the authors sought to describe all the details and steps followed for validation. However, I would like to ask a few points:

Sample:

I would like the presentation of evidence or sample calculation that indicates that the sample size is sufficient for the validation of the test, this would give greater weight to the work. Furthermore, I think that 30 players is a very small number for validation.

Metrics used

Would you like it to be clearer how the response time was analyzed in decision making? Does the system allow you to account for this response time and differentiate when the player makes a correct decision or a wrong decision? Because a player who makes a correct decision slower is much better than a player who makes a wrong decision faster. It wasn't very clear to me.

 Results and discussion:

Regarding the results, the presentation is satisfactory and very clear, however, I would like to see more results of the test itself to analyze how it can be used in the context of practice, but I am satisfied with what was presented.

 Discussion:

In addition to the preliminary data of the study, highlight its real practical implications and how it can be used at different levels of performance.

Round 2

Reviewer 2 Report (New Reviewer)

Dear authors, I congratulate you for your efforts to meet the requested revisions.

I declare that I am satisfied with the final paper.

This manuscript is a resubmission of an earlier submission. The following is a list of the peer review reports and author responses from that submission.

Round 1

Reviewer 1 Report

This study aimed to design and validated a test that examines execution time and decision-making in subcategories type of action and direction of movement. Overall, this seems to be a very interesting idea. However, the manuscript needs serious improvements before it could be considered for publication in Behavioral Sciences. Please find my comments below.  

Abstract

1) The authors should include the study’s background in the abstract. 

Introduction

2) Since this study was performed in football, it is recommended to include previous research examples in the text. It is not clear what knowledge on the execution time and decision-making in football (e.g., shooting, passing, experts vs. non-experts) are already available. I think that this is crucial to better involve the readers around the study’s topic and purposes. 

Materials and Methods

3) The Flow Diagram is not identified as a Figure in the text. Also, this image is cropped. Please, pay attention to these details. 

4) Line 148-149: Probably, the authors should include the references of those “paradigms used in cognitive neuroscience”. 

Results

5) The Tables used in the results should be reformulated. Tables should present subtitles when there are abbreviations so the reader can easily understand what it is present. The terminology should be consistent (e.g. deviation standard?). The order of the Tables in the text is confusing. We have Table 6 followed by Table 2 and then Table 8. I don’t understand. 

Discussion

6) The Discussion must be significantly improved. Overall, the authors did not discuss their results with previous literature, nor interpreted the results attained during the study. As an example, this is the only study that assessed interobserver reliability in execution time or direction of movement? Also, there is a really few relationships between the results achieved and football. What do the results mean for football practice? Finally, the authors do not refer to the limitations of the study. 

7) Please update the references in the text following the journal’s guidelines.  

Author Response

Answer